# Debridement, Antibiotic Pearls, and Retention of the Implant (DAPRI) in the Treatment of Early Periprosthetic Joint Infections: A Consecutive Series

**DOI:** 10.3390/pathogens12040605

**Published:** 2023-04-16

**Authors:** Pier Francesco Indelli, Stefano Ghirardelli, Pieralberto Valpiana, Lorenzo Bini, Michele Festini, Ferdinando Iannotti

**Affiliations:** 1Department of Orthopaedic Surgery, Stanford University School of Medicine, Redwood City, CA 94063, USA; 2Südtiroler Sanitätsbetrieb, 39042 Brixen, Italy; pieralberto.valpiana@sabes.it (P.V.); michele.festinibz@gmail.com (M.F.); 3Orthoapedic Sports Medicine, University of Toronto, Toronto, ON M5S 1B2, Canada; ghirardelli.stefano@gmail.com; 4Institute for Biomechanics, Paracelsus Medical University, Strubergasse 21, 5020 Salzburg, Austria; 5School of Medicine, University of Genova, 16132 Genova, Italy; binilorenzo@hotmail.com; 6Department of Orthopaedic and Trauma Surgery, San Paolo Hospital, 00053 Civitavecchia, Italy

**Keywords:** DAPRI, periprosthetic joint infections, PJI, total knee arthroplasty, infection, TKA, knee, hip, diagnosis, DAIR

## Abstract

Introduction: Periprosthetic joint infections (PJI) represent a devastating consequence following total joint arthroplasty (TJA). In this study, the authors describe a modified surgical technique developed to enhance the classical irrigation and debridement procedure (DAIR) to improve the possibilities of retaining an acutely infected TJA. Materials and Methods: This technique, debridement antibiotic pearls and retention of the implant (DAPRI), aims to remove the intra-articular biofilm allowing a higher and prolonged local antibiotic concentration by using calcium sulphate antibiotic-added beads in a setting of acute (<4 weeks from symptoms onset) PJI with pathogen identification. The combination of three different surgical techniques (tumor-like synovectomy, argon beam/acetic acid application and chlorhexidine gluconate brushing) aims to remove the bacterial biofilm from the implant without explanting the original hardware. Results: In total, 62 patients met the acute infection criteria (<4 weeks of symptoms); there were 57 males and five females. The patients’ average age at the time of treatment was 71 years (62–77) and the average BMI was 37 kg/m^2^. The micro-organism, always identified through synovial fluid analysis (culture, multiplex PCR or Next Generation Sequencing), was an aerobic Gram + in 76% (*S. Coag-Neg* 41%; *S. aureus* 16%), Gram—in 10% (*E. coli* 4%) and anaerobic Gram + in 4%. The DAPRI treatment was performed at an average of 3 days from symptoms onset (1–7 days). All patients underwent a 12-week course of post-operative antibiotic therapy (6 weeks I.V. and 6 weeks oral). All patients were available at the 2-year minimum FU (24–84 months). A total of 48 (77.5%) patients were infection-free at the final FU, while 14 patients underwent 2-stage revision for PJI recurrence. In total, four patients (6.4%) had a prolonged drainage from the wound after placement of the calcium sulphate beads. Conclusions: This study suggests that the DAPRI technique could represent a valid alternative to the classic DAIR procedure. The current authors do not recommend this procedure outside of the main inclusive criteria (acute scenario micro-organism identification).

## 1. Introduction

Total joint arthroplasty (TJA) represents a very successful procedure in restoring joint functionality. Between possible complications [1], infection is considered the most feared because of a high morbidity and mortality for the patient [2]. Although multiple periprosthetic joint infection (PJI) preventive protocols have been recently proposed [3], the incidence of this complication is still very high, as detected by many joint registries [4]. The timing of PJI detection is fundamental to perform any kind of implant-saving procedure. In 2018, the International Consensus Meeting (ICM) removed any distinction between early acute (post-surgical) and late acute (hematogenous) infections [5], referring to the length of symptomatology (<4 weeks) as the main determinant to classify a PJI as acute. Thanks to the ICM work, at the current time, two strict criteria have been established for orthopedic surgeons willing to perform an implant-saving procedure, namely (1) acute infection (<4 weeks of symptoms) and (2) micro-organism identification. When these criteria are not met, the success rate of any hardware-retaining procedure decreases significantly to less than 50% [6,7,8]. 

At the current time, many treatments have been proposed in the presence of a PJI. Between those, “debridement, antibiotics, and implant retention” (DAIR) has been designed to retain an infected implant [6]. Recently, the current authors proposed a novel surgical technique developed to enhance the classical DAIR procedure in order to increase its success rate. This technique, “debridement, antibiotic pearls, and retention of the implant” (DAPRI) [9,10], has the goal of improving the outcomes of the classic DAIR procedure in total joint arthroplasty, acute PJI treatment. The current study presents the outcome of the application of this technique to a consecutive series of infected total knee (TKA), total shoulder (TSA), and total hip arthroplasties (THA). 

## 2. Materials and Methods

The DAPRI technique has been designed to enhance the removal of the intra-articular biofilm, both from the synovial layer as well as from the component surfaces, allowing for a higher (above the minimum inhibitory concentrations—MIC) and prolonged, local antibiotic concentration by using calcium sulphate antibiotic-added beads in a strict setting of acute (<4 weeks from symptoms onset) PJI with pathogen identification. Due to this, two mandatory, inclusive criteria were established for the patients to enter the study group. These were (1) acute infection and (2) micro-organism identification. Only patients following primary or revision TKA, TSA, and THA were included. This retrospective multicenter study included patients from two referral hospitals (Palo Alto Veterans Affairs Health Care System, California, USA and Südtiroler Sanitätsbetrieb-SABES), Brixen, Italy between 2017 and 2021. 

The current authors followed the 2018 ICM definition [5] of acute infection: patients were included only if they had PJI symptoms (fever, chills, local erythema, elevated PJI serological markers, elevated synovial fluid markers) for less than 4 weeks from the presentation to the surgical team. Patients with a sinus tract were excluded from the study.

Micro-organism identification was performed in accordance with a multi-disciplinary team of experts (infectious disease specialists, microbiologists, internal medicine specialists, and geneticists) following the 2018 ICM guidelines [11] and adding, in non-conclusive scenarios, the use of molecular testing diagnostic technologies, such as Multiplex PCR [12] or Next Generation Sequencing (NGS) [13] (Figure 1). The use of advanced molecular testing diagnostic technologies has been shown to enhance the success rate of identifying the infecting organism, even if low-virulent [14]. All patients who underwent the DAPRI procedure were also stratified in terms of relative risk of recurrence of the infection according to the 2018 ICM guidelines [11] utilizing an iPhone application (Apple, Cupertino, CA, USA) as recommended by the algorithm published by Tan et al. [15].

Once the inclusion criteria had been met, all patients underwent surgical intervention. The DAPRI surgical technique included three steps: (1) biofilm identification; (2) biofilm removal; and (3) prevention of PJI recurrence.

(1)Biofilm Identification

Prior to the surgical approach, 50 cc of diluted (0.1%) methylene blue (40 cc saline and 10 cc of 0.5% methylene blue solution) was injected into the knee joint under sterile conditions as previously described [10] in order to stain the bacterial biofilm, making it easily identifiable after capsulotomy. After injection, the knee underwent multiple rounds of flexion and extension to facilitate the intra-articular distribution of the staining dye. An arthrocentesis was then performed to remove the excessive dye. Following a standard medial parapatellar approach and capsulotomy, blue staining of all intra-articular surfaces was identified.

(2)Biofilm removal

The highlighted soft-tissue biofilm was then removed with electrocautery.

This aggressive and radical “tumor-like” synovectomy was performed to remove the biofilm from the intra-articular lining because it was considered in contact with the infected intra-articular space. At this point, biofilm was addressed with 3 forms of aggression: thermic, mechanical, and chemical; two out of three forms of biofilm aggression have been always used in this consecutive series.


*Thermically-guided removal*


The recent literature has confirmed [16] that electrical stimulation has the capability to facilitate detachment of biofilm from orthopedic implant surfaces. Due to this, an argon beam coagulator (ConMED, Largo, FL, USA), set to 50 Watts, esd used in a painting, brush-like fashion on all visible surfaces on the femoral and tibial components once the polyethylene insert had been removed from the implant. 


*Mechanical removal*


At this point, a 2% Chlorhexidine gluconate-added brush was used as a scrubbing device on all visible implant components to mechanically remove the biofilm from the surfaces. This technique has been also supported by Tria et al. [17]. 


*Chemical removal*


Since 2019, the current authors also utilized an acetic acid, benzalkonium chloride (BZK)-based surgical lavage solution added (Bactisure, Zimmer–Biomet, Warsaw, IN, USA) as an anti-microbial solution [18]. After its application, abundant pulse irrigation with 9 L (L) of povidone iodine added saline was always performed in order to reduce the local toxicity of the acetic acid. 

(3)Prevention of PJI recurrence

When the intra-articular space was considered cleared of the biofilm presence, the wound was provisionally closed, the entire surgical team left the operating room, and the surgical drapes and contaminates instruments were removed from the surgical field. A new surgical field was then prepared, the surgical team scrubbed out and then re-entered the operating room after changing the surgical gowns; at this point, a new instruments table was prepared in a standard fashion. After wound re-opening, further irrigation of the joint was performed using saline pulse irrigation before re-implanting new modular components. At this point, 10 cc of calcium sulphate antibiotic-added beads (Stimulan, Biocomposites, Keele, UK) were prepared on the back table and placed in the joint before closure. The antibiotic that was to be added to the beads was always selected according to the antibiogram or to the preoperative molecular testing result obtained at the time of micro-organism identification. After the placement of an intra-articular hemovac, the soft tissues were closed in a standard fashion, making sure to seal the joint capsule using a Stratafix size-1 (Ethicon, Johnson & Johnson, New Brunswick, NJ, USA) suture; this step has been recommended to avoid a post-operative drainage which is a well-known complication of the use of calcium sulphate beads [19].

All patients followed an identical rehabilitative protocol, including immediate weight-bearing with crutches. Post-operative antibiotic treatment lasted for a minimum of 12 weeks, as recommended by the infectious disease specialists; a six-week course of intravenous antibiotic therapy was followed by a six-week course of oral antibiotic therapy. The success of the treatment was determined at a minimum follow-up (FU) of two years in the absence of clinical symptoms and with the presence of normal serological markers (ESR, C-reactive protein, and D-Dimer).

## 3. Results

### 3.1. Study Population

The study group included 62 patients who ultimately met the inclusion criteria. They were 57 males and 5 females. The patients’ average age at the time of treatment was 71 years (62–77) and the average BMI was 37 (32–46) kg/m^2^. The original total joint arthroplasty surgery is shown in Table 1.

The entire study group (62 patients) was also analyzed according to Tan et al. [15] in order to predict the relative risk of developing a periprosthetic joint infection at the time of the original surgery. Interestingly, all patients showed an average 28% preoperative PJI relative risk (minimum 10%–maximum 79%).

The infecting micro-organism was always identified through synovial fluid analysis (culture, multiplex PCR or Next Generation Sequencing), and was an aerobic Gram + in 76% (*S. epidermidis* 41%; *S. aureus* 16%), Gram—in 10% (*E. coli* 4%), anaerobic Gram + in 4%, and other species in 10% (Table 2). 

All DAPRI procedures were performed at an average of 3 days from symptoms onset (1–7 days) after the patient was evaluated by the multidisciplinary team to ensure that all the inclusive criteria were preoperatively met. A threshold of leukocytes >12,800 cells/μL was set in order to classify a PJI as acute. The calcium sulphate (CS) beads were prepared according to the antibiogram and/or the results of molecular diagnostics. The most used combination was represented by adding 1 g of Vancomycin and 240 mg of liquid Tobramycin (40 mg/mL) to 10 cc of CS (20 g). The beads were placed in the intra-articular space only. All patients underwent a 12-week course of post-operative antibiotic therapy under the supervision of infectious disease specialists at our institution; after 6 weeks of I.V. delivery, a 6-week course was added. The most frequently planned intravenous antibiotics were glycopeptides and cephalosporins. Other antibiotic regimens included the combination ampicillin/sulbactam or amoxicillin/clavulanic acid. The most frequently planned oral antibiotics were quinolones and combination of oral therapy.

### 3.2. Outcome

All patients were available at the 2-year minimum FU (24–84 months). In total, 48 (77.5%) patients were considered infection-free at the final FU. In fact, all serological markers evaluated (ESR, C-reactive protein, and D-Dimer) resulted normal and no PJI clinical symptoms were present. A total of 14 patients (22.5%) underwent two-stage revision for PJI recurrence before final FU. In this subgroup, the relative risk of an unsuccessful DAPRI procedure, according to the algorithm published by Tan et al. [15], was 60.29% (minimum 38%–maximum 97%). The micro-organism responsible for the re-infection was the same as the original PJI in all 14 patients. MRSA infections (9%) were resolved in 50% of the MRSA cases (three patients): 22% of patients who failed the DAPRI treatment were MRSA+. 

There were four patients (6.4%) that presented a post-operative complication related to the surgical intervention, that showed a prolonged drainage from the wound after placement of the calcium sulphate beads between five and fourteen days post-operation. In all cases, this complication was treated conservatively. All complications are shown in Table 3. 

## 4. Discussion

Debridement, antibiotic, and implant retention (DAIR) represents a very controversial surgical intervention to treat periprosthetic joint infections. The success rate reported by multiple authors, even in an acute scenario, is as low as 26% [20]. To improve this unsatisfactory outcome, the current authors developed a multi-modal surgical technique aimed to remove biofilms thermally, mechanically, and chemically from acutely infected total joint arthroplasties. At the same time, the authors used custom-made, antibiotic added, calcium sulphate re-absorbable beads to neutralize the remaining biofilm, preventing a joint re-infection. The success rate of this procedure (DAPRI) at a minimum FU of two years was 77.5%, which was considered sub-optimal but still satisfactory. 

Horriat et al. [6], in their review on the efficacy of DAIR in acute and chronic PJIs, showed that an earlier time of surgical intervention was linked to a significant increase in the success rate. The current authors suggest the use of modern, molecular testing technologies (Multiplex PCR and NGS) to improve [12,13] the timing of making the PJI diagnosis from symptoms onset. This was proposed in order to stop the biofilm development in an early stage, since it has been shown that 72 h are usually needed for the biofilm to be mature [21]. Time to positivity (TTP) of culture for PJI diagnosis has been, to the authors’ opinion, one of the main factors for the low success rate of many implant saving procedures. Tarabichi et al. [22] reported an average of 3.3 days for cultures from PJI to turn positive; *Staphylococcus epidermidis* and *C. acnes* were shown to need even more time. To lower the TTP of standard culture, DNA-/RNA-based diagnostic techniques, first developed for the fast diagnosing of SARS-CoV-1 infections, have been recently applied to diagnose musculoskeletal infections in general and PJI in particular in a timely manner. Few of them were used in the current study [12,13]. Unfortunately, the use of molecular testing technologies has increased the number of polymicrobial PJI [23], increasing the risk of false-positive cases. To date, a combination of standard culture, multiplex PCR, and meta-genomic NGS (mNGS) (Figure 1) appear to be a strategic combination to improve sensitivity and specificity [24]. To reinforce the necessity of multiple diagnostic strategies to set the time for reimplantation during the second stage of a revision following a PJI, Ludwick et al. [25] showed that 20% of their serologically negative re-implantations were still culture positive.

Multiple strategies have been applied by the current authors at the time of surgical intervention. These include the use of methylene blue to highlight the infected tissue [26] which needs to be removed in a tumor-like fashion and the application of an electrical stimulation to break the biofilm membrane to allow for a mechanical (by using a chlorhexadine gluconate-added brush) [17] and chemical (by using a acetic-acid, benzalkonium chloride added surgical lavage) removal [18]. The final use, in this technique, of calcium sulphate, antibiotic added beads, has been supported by Abosala et al. [27], who recommended their use in a PJI acute scenario. In a similar manner, the 2018 ICM study group found a consensus on the fact that the duration of symptoms, the time of PJI diagnosis, and the timing of the DAIR procedure are all related in establishing the success or failure of the procedure [8]. A major limitation of adding antibiotics to the calcium sulphate powder is represented by the fact that its use is still considerate “off-label” and a clear protocol on antibiotic selection and mixing strategy does not exist. In a recent literature review, it has been shown that vancomycin, gentamicin, and tobramycin have been used in association with calcium sulphate powder with minimal complications [28], however, little is known on the clinical use of different antibiotics. Moreover, the manufacturer of the calcium sulphate (CaSO_4_) beads used in the current study (Biocomposites, Keele, UK) does not endorse the use of antibiotics added to their local delivery system. The goal of including antibiotic doses in the package insert has the primary goal of informing how the antibiotic dose affects the bead set times. Furthermore, the elution and absorption of antibiotics from CaSO_4_ beads can be affected by the amount of antibiotic added, underlying renal dysfunction, and vascularity at the implantation site. Additionally, the combination of antibiotics could affect the elution properties of each antibiotic.

Our results confirmed that the success rate of any implant-saving procedure is strongly related to patients’ comorbidities. In fact, at the time of the surgical procedure, the preoperative relative risk of failure in our subgroup of patients (14 patients) who ultimately had a recurrence was around 60%. This confirmed the key role of a preoperative identification of risk factors for predicting periprosthetic joint infections [15]. The other factor related to the treatment failure in our series was the presence of a “difficult-to-treat” micro-organism, such as MRSA and vancomycin-resistant *Enterococcus (VRE)*. In fact, Horiat et al. [6] reported DAIR with a 66% success rate only in patients with minor comorbidities, while the rate of success dropped significantly in sicker patients. An increased risk of failure was also associated with DAIRs performed after a prolonged interval, multiple DAIRs, and antibiotic mismatches. Due to this finding, Veerman et al. [29] strongly recommended a preoperative optimization of the host immune response and special attention in preventing antibiotic mismatch. 

The current authors in this series never repeated the DAPRI procedure after a failure but they are aware that other authors [30] showed that the double-DAIR procedure with the temporary addition of high-dose antibiotic added cement beads during the first stage had infection control rates between 87% and 90%. On the other side, Lizaur-Utrilla et al. [31] showed a negative impact of prior failed DAIRs on the functional outcome of subsequent two-stage revisions. The failure of the DAIR procedure has been related [32] to multiple factors such as host-related factors (rheumatoid arthritis, old age, male sex, chronic renal failure, liver cirrhosis, and chronic obstructive pulmonary disease); implant-related factors (fracture as indication for the original surgery, cemented implants); factors related to severity of the infection (high serological CRP, a high bacterial inoculum, presence of bacteremia); and finally, causative micro-organisms (*S. aureus* and *Enterococci)*.

A recent study [33] showed a statistically significant positive linear trend, both in the hip as well as in the knee, in the incidence of methicillin-resistant *Staphylococcus aureus* (MRSA) PJIs over time as well as a statistically significant negative linear trend in the incidence of coagulase-negative *Staphylococcus* PJI over time. The authors of that study justified this finding with a gradual transition from hospital-acquired to community-acquired staphylococcal variants. In our study, MRSA infections represented only 9% of all PJI. On the other side, 28% of patients who failed the DAPRI treatment were MRSA+, confirming the difficulty of DAIR and its variants in fighting PJIs.

This study has several limitations. First, we presented the results of an inhomogeneous group of patients with different index procedures; a control group was not established. Second, the DAPRI multi-modal approach described here included the “off-label” execution of several surgical procedures for biofilm identification and removal which have minimal support in the literature of PJI treatment. Third, the authors presented the DAPRI outcome at a minimum FU of 2 years, which could be not sufficient in excluding the recurrence of the infection. On the other side, the surgical technique presented here has several advantages. Debridement is key to successful infection control of infection while antibiotic-loaded calcium sulphate use has repeatedly been demonstrated to be related to minimal complications, including potential for high local drug concentrations with significantly lower overall systemic exposure as we showed in this series. On the other hand, extended oral antibiotics following debridement with implant retention increased infection-free survivorship [28]. 

## 5. Conclusions

This study showed that the historical results of debridement, antibiotic, and implant retention (DAIR) in the treatment of periprosthetic joint infections could be improved adding a multi-modal surgical approach. Unfortunately, the satisfactory success rate shown in this study requires micro-organism identification in a strict, timely fashion which represents a hard achievement in many institutions since the number of culture-negative PJIs is increasing worldwide. 

## Figures and Tables

**Figure 1 pathogens-12-00605-f001:**
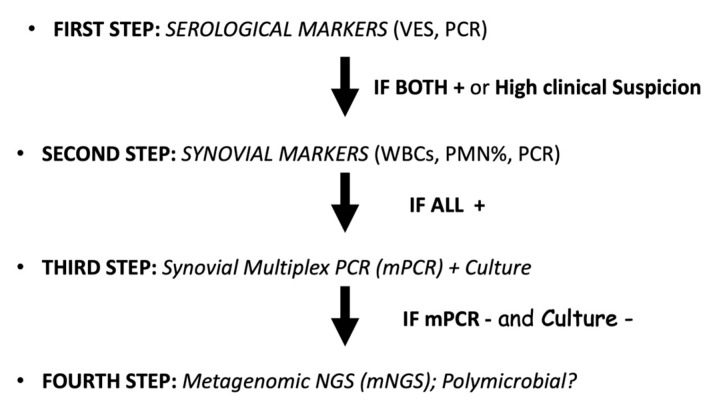
Periprosthetic joint infections (PJI) diagnostic algorithm. ESR: erythrocyte sedimentation rate; CRP: C-reactive protein; WBC: white blood cell; PMN: polymorphonuclear cell; mPCR: multiplex PCR; NGS: Next Generation Sequencing; mNGS: metagenomic NGS.

**Table 1 pathogens-12-00605-t001:** Original total joint arthroplasty (TJA) surgery.

Primary TKA	Revision TKA	Primary THA	Revision THA	Primary TSA	Revision TSA
26 patients	11 patients	13 patients	6 patients	6 patients	No

TKA: Total knee arthroplasty; THA: Total hip arthroplasty; TSA: Total shoulder arthroplasty.

**Table 2 pathogens-12-00605-t002:** Infecting micro-organisms.

Bacterial Group	Incidence	Organisms	Prevalent Organism
Aerobic Gram +	76%	*S. epidermidis* (41%), *Streptococcus* sp. (10%); MRSA (9%); MSSA (7%); *S. Lugdunensis* 5%, *S. hominis* 4%	*Staphylococcus epidermidis*
Gram −	10%	*Escherichia coli* (4%); *Enterobacter* (3%); *Enterobacter cloacae* (1.5%); *Proteus mirabilis* (1.5%)	*Escherichia coli*
Other pathogen	10%	*Enterococcus faecalis* (5.5%; VRE 1.5%); *Candida albicans* (1.5%); *Corynebacterium striatum* (1.5%)	*Enterococcus faecalis*
Anaerobic Gram +	4%	*Cutibacterium acnes (4%)*	*Cutibacterium acnes*

MSSA: Meticillin-sensitive *Staphylococcus aureus*; MRSA: Methicillin-resistant *Staphylococcus aureus*; VRE (Vancomycin-resistant *Enterococcus*).

**Table 3 pathogens-12-00605-t003:** Complications.

Persistent Drainage	Hypercalcemia	PJI Recurrence	Heterotopic Ossifications
4 patients (knees) (6.4%)	None	4 patients (22.5%)	1 patient (hip) (1.6%)

## Data Availability

Data supporting reported results can be requested to the senior author (P.F.I.).

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
