# Peer review of "Debridement, Antibiotic Pearls, and Retention of the Implant (DAPRI) in the Treatment of Early Periprosthetic Joint Infections: A Consecutive Series"

_pathogens, 2023, doi:10.3390/pathogens12040605_

Round 1

Reviewer 1 Report

This is a very well written manuscript. If more total cases and more female cases can be collected, that should be better. The complication seems too simple, I hope the authors can add more detail on the postoperative complications to the manuscript.

Author Response

Thank you for your review of our work. We really appreciated your compliments and recommendations. We revised the complication section of the manuscript, focusing on the wound drainage issue (line 218-227; Table 3). 

Reviewer 2 Report

This is an interesting study on an important health issue (PJI), where optimal management is not yet clear. Therefore, the strategy that authors attempted in this study warrants further evaluation.

Overall, the manuscript needs improvements in its scientific quality of language, especially in the Discussion section. Please take care to improve the scientific soundness of the discussion.

Title: authors could consider adding the word “early” in treatment.

Figure 1. please correct to “clinical suspicion”

Methods: please describe the setting and time of the study; currently it’s not clear. Also, the authors mention Ethical approval from the IRB of SABES. Please define SABES and mention whether all study participants were in this center.

Line 81: please correct to “surgical intervention”.

Table 2. I’d suggest presenting the microorganisms in descending incidence.

Line 153: this is not prophylaxis but treatment, since the patients had PJI.

­Line 188: please describe all antibiotic regimens used for treatment, including any empirical treatment given.

Line 198: this is not clear to me; why do authors mention a different study in relation to the outcome of their study? This should be part of the discussion. In addition, it would be useful to do a comparison of characteristics between patients with successful outcome vs the 14 patients who required an intervention at follow-up.

Line 252: please correct “addiction”.

Author Response

Reviewer 2

Thank you for your extensive and exhaustive review of our work. We really appreciated your compliments and recommendations.

Title: We added the word “early” in the manuscript title.

Figure 1: We corrected the figure legend including the “clinical suspicion” criteria (Line 98).

Methods: 

We described the setting and time of the study (Line 77-80; Line 346-349)
Table 2: 

We presented the microorganisms in descending incidence.

Line 164: The word “treatment” was included, and the word “prophylaxis” removed.

Line 202-204: The antibiotic regimens were described.

Line 213 and Lines 288-292. We clarified the citation in Line 213. We also made a comparison of patient characteristics between patients with a successful outcome and patients who failed the surgical intervention (Lines 288-292). 

Line 271: The word “adding” was inserted. 

Thank you again.

Round 2

Reviewer 2 Report

Thank you for addressing my comments. Please see some additional comments below. Clarity and improvements in scientific language are requested.

Please add more details in Table 2 and in accordance to the text. For example, text mentions 16% were S.aureus but this is omitted from the table where only MRSA (9%) is mentioned; the percentage of E.coli is mentioned but not of C.albicans. Ideally, add ALL pathogens and their percentages. Authors should take care to be clear and detailed when presenting such important information.  Additionally, authors mention only SOME resistance patters (eg.MRSA) and then in the discussion they add VRE. Please clarify and decide whether resistances will be mentioned in the microbiology of infections or not.

Table 3. Please present data in a similar manner, PJI recurrence is mentioned only in % whereas other complications are in n(%). 

Line 210: this comparison is irrelevant here; please remove or discuss further in the Discussion section.

Line 277: the authors here mention a risk rate(?) which wasn't mention in the results. Please clarify. If authors wish to perform a risk analysis please do so, otherwise avoid ambiguous statements.

Author Response

Answers to Reviewer(s)' Comments to Author:

Reviewer 2

Thank you again for your extensive and exhaustive review of our work. We really appreciated your compliments and recommendations.

TABLE 2: Table 2 underwent major revision (lines 191-197). We included ALL pathogens and their percentages. We also clarified the incidence of MRSA infections and VRE infections (resistance patterns) both in table 2 as well as in Outcome section (230-232) and the Discussion section (304-306).

TABLE 3: Table 3 underwent major revision (lines 238-239). We included the percentages of PJI recurrence and other complications.

Line 210-Line 277: We addressed the issue of the risk analysis assessment, which was not clear in the previous submission. The current authors performed a risk analysis (Materials and Methods, Lines 90-93) according to the Tan et al. [15] algorithm. The results of this risk analysis assessment have been reported in lines 187-189 and 228-230. We discussed this relationship in the Discussion section in lines 300-304.